# Testing the psychometric properties of the Swedish version of the EPOCH measure of adolescent well-being

**Mia M. Maurer** \* , **Daiva Daukantaitė, Eva Hoff**

Department of Psychology, Lund University, Lund, Sweden

\* mia.maurer@psy.lu.se

## Abstract

The EPOCH Measure of Adolescent Well-being measures five positive indicators of the well-being of adolescents: engagement, perseverance, optimism, connectedness and happiness. This five-factor structure along with other indicators of validity and reliability were supported for the original English version and the Chinese version. In this study, we tested the psychometric properties of the Swedish version of the EPOCH with a sample ($n$ = 846) of Swedish high school adolescents aged 16–21 years ($M_{age}$ = 18, $SD$ = .85). The participants answered a questionnaire containing the EPOCH, Coping Self-Efficacy Scale, and 21-item Depression, Anxiety, and Stress Scale (DASS-21). A confirmatory factor analysis supported a the five-factor, inter-correlated model. The internal consistency was good for all the EPOCH subscales (Cronbach's α = .76–.88, McDonald's ω = .77 –.88). The criterion validity was established by replicating correlations between the five EPOCH subscales and positive (coping self-efficacy) and negative (DASS-21) aspects of well-being. This study shows that the Swedish version of the EPOCH is suitable for assessing multiple dimensions of adolescent well-being.

## Introduction

Positive mental health is conceptualized not only as a lack of mental health problems such as anxiety, stress and depression, but also by the presence of positive psychological well-being indicators such as perseverance, optimism and happiness [1,2]. Measuring adolescent well-being using a multidimensional approach—that is, encompassing multiple positive indicators —is important, given the multitude of concepts relevant for young people's well-being [3]. Furthermore, as well-being comprises both hedonic ("feeling good") and eudaimonic ("functioning well" [4]) aspects, it is important to take both into consideration when measuring well-being. In Sweden, a valid multidimensional measure of adolescent well-being does not yet exist. This study therefore aims to validate the Swedish version of the EPOCH Measure of Adolescent Well-being (hereafter, EPOCH) [2]. The EPOCH measure [2] has the theoretical strength of being both a multidimensional measure (consisting of five factors of well-being: engagement, perseverance, optimism, connectedness and happiness) and a measure of both the hedonic and eudaimonic aspects of well-being, discussed more next. Furthermore, the

**Data Availability Statement:** Data for this study cannot be made publicly available because the study participants provided informed consent only to the original study's aims. Furthermore, it was

made sure that the participants knew that their data would not be publicly shared. However, it is possible to request approval for a copy of the de-identified dataset from the Data Protection Office at Lund University, Box 117, 221 00 Lund, Sweden, telephone: +46 46 222 0000, dataskyddsombud@lu.se (please quote project number 2019-06552).

**Funding:** This work was supported by the Crafoord Foundation (20180643; https://www.crafoord.se/en/ ) and the Thora Ohlsson Foundation (https://www.stiftelsemedel.se/thora-ohlssons-stiftelse/) to DD. The funders had no role in study design, data collection and analysis, decision to publish, or preparation of the manuscript.

**Competing interests:** The authors have declared no competing interests exist.

EPOCH measure is designed to operationalize PERMA theory of well-being [5] for adolescents. The PERMA model (discussed in the next section) is the main model used in positive educational studies; accordingly, having a valid measure of it will be important particularly in the positive educational research context.

## Well-being theories and the EPOCH measure of adolescent well-being

Traditionally, well-being has been approached from either the hedonic or the eudaimonic viewpoint [4,6], although debate about the validity of this distinction also exists [7,8].Hedonic well-being broadly refers to emotional well-being, including frequent positive emotions and infrequent negative emotions, as well as a sense of life satisfaction [4]. Student life satisfaction is commonly measured with the Students' Life Satisfaction Scale [9], which measures students' context-free estimation of their life satisfaction as a whole. This scale has been used in both clinical and non-clinical studies with student samples ranging in age from 8 to 18 years [10,11]; however, it is not a multidimensional measure. Separating the different dimensions of well-being can help to recognize areas in which students particularly flourish or flounder, and can suggest further efforts for different types of intervention in areas that are lacking [3]. For instance, the Multidimensional Students' Life Satisfaction Scale [12,13] for 8- to 18-year-olds measures student life satisfaction across five important domains: school, family, friends, self and living environment.

Eudaimonic well-being refers to well-being resources and positive functioning, such as a sense of meaning in life, authenticity, personal growth, autonomy and positive relationships [4,6,14]. Although thorough consideration of the debate about the validity of distinguishing between various types of well-being is beyond the scope of this article, some researchers advocate for the distinction of hedonia from eudaimonia [e.g. 4,6,15], whereas others oppose this distinction [e.g. 7,8]. For instance, Longo et al. [16] studied the factor structure of Huppert and So's [17] 10 well-being indicators—happiness, emotional stability, vitality, resilience (aspects of "feeling good", or hedonia), competence, engagement, meaning, optimism, positive relationships and self-esteem (aspects of positive functioning, or eudaimonia)—in two different studies and found that instead of two distinct factors, the items loaded onto a single higher-order well-being factor. However, two factor analytic studies, Linley et al. [18] and Joshanloo [19], found evidence for distinct factors. In any case, a full account of well-being arguably requires consideration of both the hedonic and eudaimonic aspects. One theory encompassing both aspects is the PERMA theory of well-being [5], which approaches well-being as a multidimensional concept consisting of positive emotion, engagement, positive relationships, meaning and achievement.

The PERMA model is used commonly as a theoretical framework for school applications of positive psychology, called positive education [20], with an emphasis on providing a multidimensional view of well-being for school youth [20]. Positive education involves bringing the science of well-being into schools through well-being lessons or through influencing the school culture in other ways [20]. The PERMA profiler, a measure developed for the PERMA model, was validated in its original development study with an international sample recruited online [21] and in Turkish [22], Australian [23], Italian [24] and Indonesian [25] samples. Although some results were mixed or unsatisfactory [e.g. 26], generally the results have supported the five-factor model with high internal and test-retest reliability, factor structure and construct validity [21,24,25].

Since the PERMA model is the main theoretical framework used in positive educational studies [20,27], it is important to validate a measure based on it for adolescents. Kern et al. [2] wanted to translate the PERMA model to better suit adolescents and therefore operationalized

a measure of optimal adolescent functioning that reflects both the attitudes and characteristics related to optimal outcomes for adolescents: the EPOCH Measure of Adolescent Well-being. The EPOCH consists of five subcomponents, including engagement, perseverance, optimism, connectedness and happiness. The subcomponents reflect the PERMA categories, albeit using different language, and relate to similar outcomes according to Kern et al. [2]. The motivation for including characteristics (i.e. personality traits such as perseverance) in a definition of well-being is practical: the EPOCH was developed to be used in particular positive educational intervention contexts, in which different aspects relevant for optimal adolescent functioning should be assessed in order to develop, improve upon, and evaluate interventions. Kern et al. [2] claim that using the EPOCH categories can help in developing more targeted interventions and show where adolescents might be lacking in particular. Although not a diagnostic tool, the EPOCH can aid in the development and assessment of interventions [2] aiming to promote other positive psychological constructs, such as optimism and perseverance, on top of happiness.

In the EPOCH, *engagement* refers to the ability to become absorbed in what one is doing, with its most intense form referring to a sense of 'flow' as defined by Csikszentmihalyi [28]– a state of complete absorption is what one is doing with the loss of a sense of time and self. *Perseverance* is the ability to keep pursuing one's activities and goals even in the case of setbacks on one's way. Similar concepts include grit, defined by Duckworth et al. [29] as 'passion and perseverance to long-term goals'. *Optimism* refers to a sense of hopefulness about the future along with a tendency to have an explanatory style in which positive events are attributed to global and internal causes (i.e. one feels a sense of agency in making positive events possible), whereas negative events are attributed to specific and external events (i.e. one feels not directly accountable for negative events and these events are just temporary occurrences) [2,30]. *Connectedness* refers to one's sense of being cared for as a person by others through positive interpersonal relationships. Finally, *happiness* refers to the common occurrence of positive emotionality, such as joy and a love of life [2].

Support was found for this five-factor structure of the EPOCH, both for the English [2] and Chinese versions [31]. Kern and colleagues [2] evaluated the factor structure of the original English EPOCH scale when testing the psychometric properties of the scale via 10 different studies with over 4,000 adolescents from the USA and Australia. The five EPOCH subscale scores were found to have high internal consistency (happiness had the highest at $\alpha = .87$, and engagement the lowest at $\alpha = .76$). The subscale scores were not correlated with age and gender but were weakly to strongly correlated with other well-being indicators (significant correlations ranged from $r = .12$ for optimism and autonomy to $r = .83$ for happiness and life satisfaction) and negatively correlated with mental health symptoms (ranging from $r = -.29$ for optimism and anxiety to $r = -.53$ for happiness and depression). Furthermore, Zeng and Kern [31] tested the Chinese version of the EPOCH with a sample of 17,854 adolescents from different regions in China and found that the five-factor structure was supported with good model fit. The subscale scores were internally consistent, but its test-retest reliability was low (with a range from $r = .12$ to $r = .21$). The criterion validity was good: the subscale scores were consistently positively correlated with various well-being indicators (ranging from $r = .16$ between happiness and growth mindset to $r = .57$ between happiness and coping) and negatively with mental health symptoms (ranging from zero correlation between engagement and anxiety to $r = -.25$ between happiness and depression). The correlations with mental health symptoms were slightly weaker than were those with the positive indicators. In general, engagement showed the weakest correlations with well-being indicators as compared to the other subscales, while optimism and happiness showed the strongest correlations.

## Current study

In this study, we investigated the psychometric properties of the Swedish version of the EPOCH Measure of Adolescent Well-being. To establish its criterion validity, we evaluated its correlations with coping self-efficacy, a positive indicator of well-being [32], and anxiety, stress and depression symptoms, all of which are negative indicators [33].

In line with earlier studies [2,31], we expect the Swedish version of the EPOCH to have a five-factor structure with reliable subscales. Furthermore, we expect to find a positive association between the EPOCH and coping self-efficacy and a negative association between the EPOCH and mental health symptoms. More specifically, we expect that the engagement subscale will show the weakest association with well-being and mental health indicators (coping self-efficacy and DASS-21, respectively), while the happiness subscale will show the strongest association (thus replicating [2,31]). Finally, based on previous research [2], we also expect the EPOCH to have weak relationships with gender.

## Materials and methods

### Participants

The study was administered online on the school's webpage. Interested students could click on the study link. Altogether, 1212 students opened the study link and out of those 852 (70.3%) responded. We excluded six participants because they did not correctly answer two or more control questions randomly placed in the survey to assess attentiveness, leaving a total of 846 participants. The participants came from three schools matched in terms of study programmes, sizes and reputation for having highly motivated students in three Swedish cities. Participants ranged in age from 16 to 21 years, with a mean age of 18 years ($SD$ = .85). The majority of the participants ($n$ = 555, 65.6%) were female, 286 (33.8%) were male, and 5 (0.6%) reported 'other'. Most participants were born in Sweden ($n$ = 722, 85.3%), with either one or both parents being born in Sweden ($n$ = 575, 68%). All the students were enrolled in a Swedish school in a Swedish-speaking program. Therefore, their Swedish was considered fluent for the purposes of validating the EPOCH.

### Procedure

The data were collected during the baseline assessment of a larger project assessing students' mental health during the COVID-19 pandemic. The battery of measures was distributed through the websites of three schools in three Swedish cities in May 2020 to be completed at home, since all schools were closed due to the pandemic. The school administration agreed to participate in the study. Ethical approval for the study was obtained from the Swedish Ethical Review Authority. All students were over age 15, meaning that, according to Swedish ethical guidelines, they could give their own informed consent and no parental informed consent was required. The students were given information about the study, which emphasized that their responses would be kept anonymous and confidential. All participants provided informed consent before participating in this study by selecting a box on an online form, indicating their understanding of the nature of the study and that they agreed to participate.

### Measures

**EPOCH measure of adolescent well-being.** All participants responded to the 20-item EPOCH, which contains subscales of engagement (e.g. '*I get completely absorbed in what I am doing*'), perseverance (e.g. '*I finish whatever I begin*'), optimism (e.g. '*I think good things are going to happen to me*'), connectedness (e.g. '*When I have a problem, I have someone who will*

*be there for me*') and happiness (e.g. '*I love life*'). Each item is rated on a 5-point Likert scale ranging from 1 '*Almost never*' to 5 '*Almost always*'. A back-translation process was used to translate the scale into Swedish [34]. The scale was translated into Swedish by two native Swedish speakers, and then back-translated into English. Another native English speaker checked the back-translation for similarity of meaning with the original scale. There was high level of agreement in the similarity of meanings of all items in the scale. Kern et al. [2] and Zeng and Kern [31] previously found that all subscales had good internal consistency, with Cronbach's α values ranging from 0.78 to 0.89 [31]. The internal consistency of the EPOCH for this study is reported in the Results section.

**Coping self-efficacy scale.** The Coping Self-Efficacy Scale (CSE) [32] consists of 13 items in three subscales: using problem-focused coping (e.g. '*Think about one part of the problem at a time*'), stopping unpleasant emotions and thoughts (e.g. '*Take your mind off unpleasant thoughts*') and getting support from friends and family (e.g. '*Get emotional support from friends and family*'). The items are rated on an 11-point Likert scale ranging from 0 ('*cannot do at all*') to 10 ('*certain can do*'). In this study, the internal consistency of the whole scale was high (α = 0.89), as was that of the subscales: using problem-focused coping (α = 0.85), stopping unpleasant emotions and thoughts (α = 0.88), and getting support from friends and family (α = 0.78).

**Depression, anxiety and stress scale.** The DASS-21 [33] consists of 21 items in three subscales: depression, which measures dysphoria, self-depreciation, lack of interest and hopelessness (e.g. '*I felt that I had nothing to look forward to*'); anxiety, which measures affective experience of anxiety, autonomic arousal and muscle effects (e.g. '*I felt I was close to panic*'); and stress, measuring chronic arousal (e.g. '*I found it hard to wind down*'). The items are rated on a 4-point Likert scale ranging from 0 ('*did not apply to me at all*') to 3 ('*applied to me very much, or most of the time*'). In this study, the internal consistency of the whole scale was high (α = 0.91), while the subscales also satisfactory to high internal consistencies: depression (α = 0.89), anxiety (α = 0.73) and stress (α = 0.81).

## Data analyses

The factor structure of the Swedish version of the EPOCH was tested using confirmatory factor analysis (CFA) with maximum likelihood estimation and robust standard errors. These analyses were conducted using Mplus version 8 [35]. Replicating Kern et al. [2], in addition to a five-factor model with inter-related latent variables corresponding to each EPOCH subscale, we estimated a one-factor model (i.e. all items loaded onto a single well-being factor) and a second-order model (i.e. items loading onto five factors, which in turn load onto a single overarching well-being latent construct) and compared these latter two models to the first one. The goodness of fit of the models was evaluated using the $\chi^2$ statistic, where a nonsignificant value represents an acceptable fit. Because of the chi-square test's sensitivity to sample size [36], we also computed several approximate fit indices with conventional cutoffs: the root mean square error of approximation (RMSEA) with 95% confidence interval, Tucker-Lewis Index (TLI), comparative fit index (CFI) and standardized root mean square residual (SRMR). Acceptable fit standards are as follows: TLI and CFI $\geq$ 0.90, RMSEA $\leq$0.08 and SRMR $\leq$0.10 [37]. To test measurement invariance across genders, we compared increasingly constrained models to a less constrained model, focusing on the change in CFI (ΔCFI) and RMSEA (ΔRMSEA). We considered ΔCFI $\leq$ 0.010 and ΔRMSEA $\leq$ 0.015 [38,39] as indicative of the invariance assumption holding. The internal consistency was estimated with the use of Cronbach's alpha and McDonald's ω levels, with a criterion value of >.07 for acceptable consistency [40]. Cronbach's alpha is a good measure of internal consistency, particularly with multidimensional scales (i.e. those that measure different latent constructs), whereas the McDonald's ω is better suited measuring the internal consistency of

unidimensional scales (i.e. those that measure the same latent construct; [41]). If items are unidimensional, Cronbach's alpha gives a reliable measure of internal consistency only once all items have equal covariance with the true score, which is seldom the case [41]. Since we are testing both one-factor and five-factor models, we decided to report both indicators of internal consistency. The criterion validity was estimated with Pearson product moment correlations between the EPOCH total score and subscale scores and the indicators of coping self-efficacy (i.e. CSE) [32] and mental health symptoms (i.e. DASS-21) [33].

## Results

### Preliminary analyses

The data included some missing values (altogether < .01%), but an analysis of missing values with Little's missing completely at random (MCAR) test yielded a non-significant result, $\chi^2$ (5595) = 5614.25, $p$ = 0.46. Thus, the missing values were not related to any variables in our study. Full information maximum likelihood (FIML) was used to handle the missing data.

### Construct validity of the EPOCH

The five-factor model showed acceptable model fit ($\chi^2$(160) = 818.72, $p$ < .001; CFI = 0.917, TLI = 0.902, RMSEA = 0.070 [CI = 0.065, 0.075], SRMR = 0.044) and fit better than the 1-factor model [$\chi^2$(170) = 2530.24, $p$ < .001; CFI = 0.703, TLI = 0.668, RMSEA = 0.128 CI [0.124, 0.133], SRMR = 0.091; $\Delta\chi^2$(10) = 1711.52, $p$ < .001;) and second-order factor model ($\chi^2$(165) = 871.78, $p$ < .001; CFI = 0.911, TLI = 0.898, RMSEA = 0.071 (CI = 0.067, 0.076), SRMR = 0.050; $\Delta\chi^2$(5) = 53.06, $p$ < .001), even though the latter model's fit was also acceptable. The CFI and TLI were both above the desired cut-off of .9 (TLI was about .9 for the second-order model), the RMSEA was below the critical .08 threshold, and the SRMR was below .09 [37] for both the five-factor and the second-order model. Therefore, both total scale scores and subscale scores are used in analyses. All items had loadings of >.40 onto their respective factors and were significant at $p$ < .01 in the five-factor model. Scalar invariance was supported for gender (see Table 1), indicating that the five-factor model is sufficient and that mean values are directly comparable across the genders. The five-factor solution of the EPOCH is presented in Fig 1.

### Internal consistency

The total EPOCH scale had high internal consistency (Cronbach's $\alpha$ = 0.91, McDonald's $\omega$ = 0.92). The subscales likewise all had high or acceptable consistencies: *engagement* ($\alpha$ = 0.79, McDonald's $\omega$ = 0.80), *perseverance* ($\alpha$ = 0.76, McDonald's $\omega$ = 0.77), *optimism* ($\alpha$ = 0.83, McDonald's $\omega$ = 0.84), *connectedness* ($\alpha$ = 0.74, McDonald's $\omega$ = 0.75) and *happiness* ($\alpha$ = 0.88, McDonald's $\omega$ = 0.89).

### Criterion validity

The intercorrelations were estimated with Pearson product moment correlation coefficients between the total and subscales of the EPOCH and positive and negative well-being indicators

**Table 1. Comparing configural, metric and scalar invariance across gender.**

| Invariance | df | AIC | BIC | $\chi^2$ | $\Delta\chi^2$ | p | CFI | RMSEA | $\Delta$CFI | $\Delta$RMSEA |
|---|---|---|---|---|---|---|---|---|---|---|
| Configural | 320 | 40743.66 | 41406.50 | 986.22 | | | .916 | .070 | | |
| Metric | 335 | 40747.75 | 41339.58 | 1020.31 | 34.09 | .003 | .914 | .070 | .002 | .000 |
| Scalar | 350 | 40789.69 | 41307.50 | 1089.25 | 68.94 | < .001 | .907 | .071 | .007 | .001 |

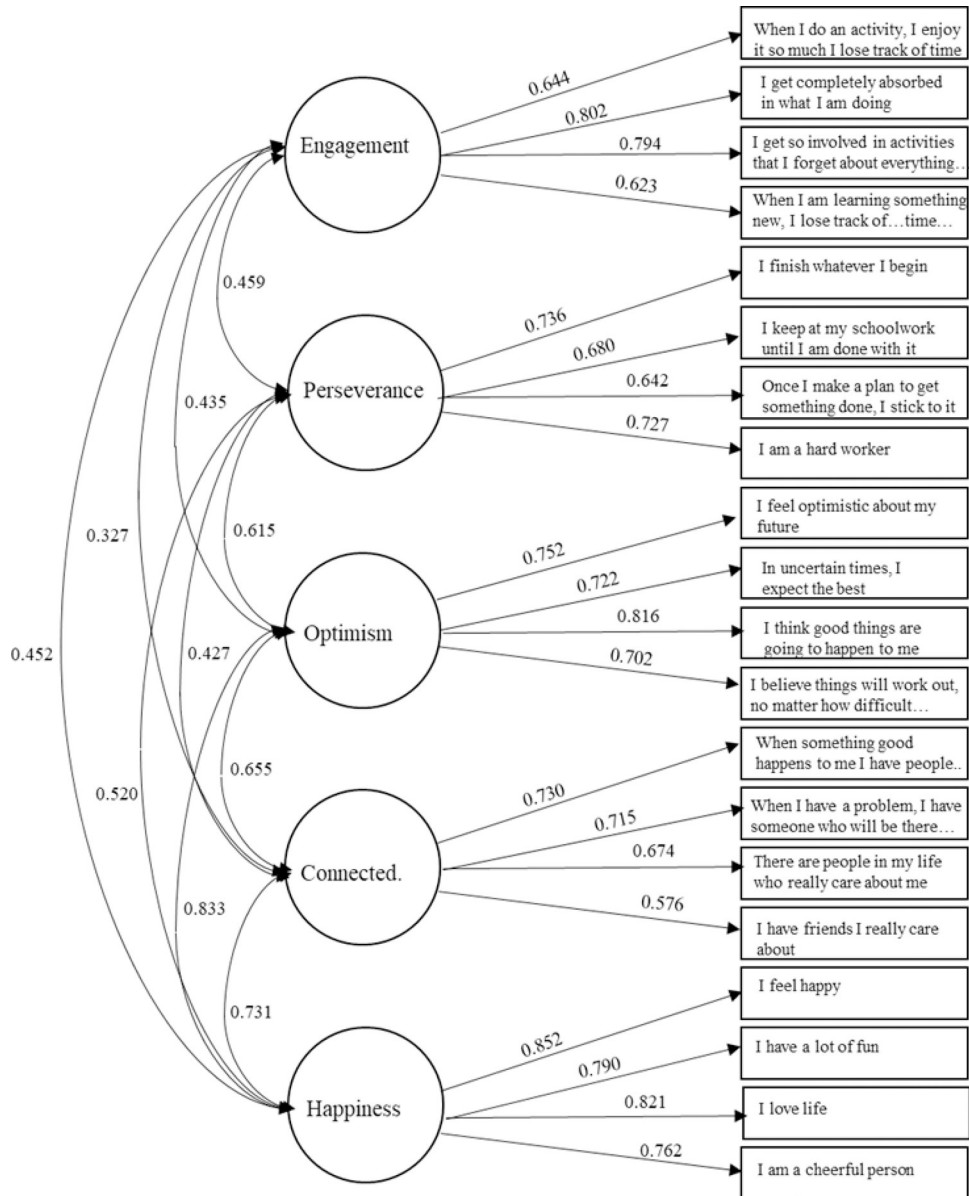

**Fig 1. Confirmatory factor analysis of the EPOCH item loadings onto their subcomponents, showing the standardized factor loadings and latent correlations (some items are abbreviated).**

(CSE and DASS-21, respectively, see Table 2). Moreover, all item intercorrelations were estimated (see Table 3).

All intercorrelations were significant at $p < .001$ (see Table 4). As expected, the total EPOCH score had a strong positive correlation with the CSE ($r = 0.68$) and a strong negative correlation with the DASS-21 ($r = -0.56$). The engagement subscale showed only a moderate positive correlation with the CSE ($r = 0.32$) and a weak negative correlation with the DASS-21 ($r = -0.17$). Engagement showed weak negative correlations with all the DASS-21 subscales and weak positive correlations with all the CSE subscales (see Table 2). Perseverance had a moderate positive correlation with the CSE ($r = 0.46$) and a moderate negative correlation with the DASS-21 ($r = -0.34$), as expected. Perseverance likewise showed weak to moderate

**Table 2. Descriptive statistics for the EPOCH subscales and intercorrelations between the EPOCH subscales and overall scale ($N$ = 846).**

|  | Engagement | Perseverance | Optimism | Connectedness | Happiness | Overall |
|---|---|---|---|---|---|---|
| E | — |  |  |  |  | .64 |
| P | .37 | — |  |  |  | .69 |
| O | .37 | .49 | — |  |  | .84 |
| C | .27 | .34 | .52 | — |  | .71 |
| H | .38 | .43 | .72 | .61 | — | .85 |
| M (SD) | 3.08 (.084) | 3.74 (0.78) | 3.65 (0.89) | 4.41 (0.69) | 3.69 (0.91) | 3.72(0.62) |

E = Engagement; P = Perseverance; O = Optimism; C = Connectedness; H = Happiness.

All correlations were significant at $p < .01$.

negative correlations with the DASS-21 subscales, and moderate to strong positive correlations with the CSE subscales. Optimism, by contrast, had a strong positive correlation with the CSE ($r = 0.66$) and a strong negative correlation with the DASS-21 ($r = -0.56$). Optimism also had moderate to strong negative correlations with the DASS-21 subscales, and a strong positive

**Table 3. EPOCH scale item intercorrelations and item descriptives ($N$ = 846).**

| | M(SD) | 1. | 2. | 3. | 4. | 5. | 6. | 7. | 8. | 9. | 10. | 11. | 12. | 13. | 14. | 15. | 16. | 17. | 18. | 19. | 20. |
|---|---|---|---|---|---|---|---|---|---|---|---|---|---|---|---|---|---|---|---|---|---|
| **1.** | 4.43(.92) | | | | | | | | | | | | | | | | | | | | |
| **2.** | 3.93(.92) | .23 | | | | | | | | | | | | | | | | | | | |
| **3.** | 3.74(1.14) | .37 | .44 | | | | | | | | | | | | | | | | | | |
| **4.** | 3.62(1.02) | .41 | .35 | .57 | | | | | | | | | | | | | | | | | |
| **5.** | 3.73(1.15) | .19 | .15 | .21 | .21 | | | | | | | | | | | | | | | | |
| **6.** | 3.69(.97) | .47 | .36 | .52 | .72 | .31 | | | | | | | | | | | | | | | |
| **7.** | 3.07(1.07) | .18 | .29 | .31 | .30 | .47 | .40 | | | | | | | | | | | | | | |
| **8.** | 3.55(1.20) | .38 | .34 | .57 | .73 | .26 | .65 | .36 | | | | | | | | | | | | | |
| **9.** | 3.24(1.22) | .14 | .52 | .31 | .26 | .14 | .26 | .26 | .27 | | | | | | | | | | | | |
| **10.** | 4.01(1.13) | .52 | .26 | .37 | .50 | .19 | .49 | .20 | .48 | .27 | | | | | | | | | | | |
| **11.** | 2.57(1.09) | .14 | .17 | .22 | .19 | .48 | .30 | .59 | .24 | .23 | .16 | | | | | | | | | | |
| **12.** | 2.95(.99) | .13 | .23 | .23 | .20 | .37 | .27 | .44 | .26 | .30 | .17 | .50 | | | | | | | | | |
| **13.** | 3.34(1.05) | .28 | .33 | .48 | .49 | .20 | .48 | .29 | .52 | .21 | .33 | .25 | .30 | | | | | | | | |
| **14.** | 4.65(.72) | .43 | .20 | .31 | .39 | .12 | .36 | .17 | .40 | .15 | .53 | .09* | .12 | .31 | | | | | | | |
| **15.** | 3.97(.98) | .36 | .42 | .68 | .58 | .17 | .54 | .27 | .64 | .29 | .43 | .17 | .22 | .57 | .44 | | | | | | |
| **16.** | 4.55(.83) | .49 | .19 | .26 | .29 | .19 | .38 | .15 | .29 | .13 | .32 | .17 | .14 | .24 | .37 | .31 | | | | | |
| **17.** | 3.74(.92) | .22 | .49 | .31 | .35 | .17 | .35 | .27 | .31 | .36 | .22 | .21 | .25 | .30 | .19 | .36 | .30 | | | | |
| **18.** | 3.55(1.12) | .31 | .32 | .47 | .40 | .24 | .44 | .30 | .50 | .20 | .33 | .22 | .27 | .70 | .30 | .58 | .29 | .38 | | | |
| **19.** | 4.07(.93) | .16 | .52 | .34 | .26 | .17 | .26 | .30 | .28 | .50 | .21 | .18 | .27 | .21 | .17 | .36 | .16 | .44 | .26 | | |
| **20.** | 3.90(1.00) | .39 | .30 | .49 | .67 | .18 | .61 | .29 | .66 | .19 | .42 | .19 | .19 | .48 | .39 | .58 | .36 | .31 | .47 | .29 | |

1. = When something good happens to me, I have people who I like to share the good news with; 2. = I finish whatever I begin; 3. = I am optimistic about my future; 4. = I feel happy; 5. = When I do an activity, I enjoy it so much that I lose track of time; 6. = I have a lot of fun; 7. = I get completely absorbed in what I am doing; 8. = I love life; 9. = I keep at my schoolwork until I am done with it; 10. = When I have a problem, I have someone who will be there for me; 11. = I get so involved in activities that I forget about everything else; 12. = When I am learning something new, I lose track of how much time has passed; 13. = In uncertain times, I expect the best; 14. = There are people in my life who really care about me; 15. = I think good things are going to happen to me; 16. = I have friends that I really care about; 17. = Once I make a plan to get something done, I stick to it; 18. = I believe that things will work out, no matter how difficult they seem; 19. = I am a hard worker; 20. = I am a cheerful person.

* $< .05$. All other correlations are significant at $p < .01$.

Table 4. **Intercorrelations between the EPOCH subscales with well-being indicators (N = 842–846).**

| | EPOCH | | | | | |
| --- | --- | --- | --- | --- | --- | --- |
| | **Engagement** | **Perseverance** | **Optimism** | **Connectedness** | **Happiness** | **Overall** |
| **Well-being indicators:** | | | | | | |
| Coping self-efficacy, total | 0.32 | 0.46 | 0.66 | 0.48 | 0.61 | 0.68 |
| Problem-focused c. | 0.28 | 0.51 | 0.55 | 0.33 | 0.43 | 0.56 |
| Emotion-focused c. | 0.25 | 0.28 | 0.54 | 0.28 | 0.51 | 0.50 |
| Social support | 0.24 | 0.30 | 0.53 | 0.63 | 0.58 | 0.61 |
| DASS total | -0.17 | -0.34 | -0.56 | -0.38 | -0.63 | -0.56 |
| Depression | -0.22 | -0.41 | -0.59 | -0.50 | -0.70 | -0.65 |
| Anxiety | -0.10 | -0.22 | -0.41 | -0.26 | -0.44 | -0.39 |
| Stress | -0.12 | -0.23 | -0.44 | -0.22 | -0.46 | -0.40 |

Coping SE = Coping self-efficacy; Problem-focused c = using problem-focused coping; Emotion-focused c = stopping unpleasant emotions and thoughts; Social support = getting support from friends and family.

Note: All the correlations with the W-B indicators were significant at $p < .001$, not indicated here.

correlation with the CSE subscales. Connectedness showed moderate positive correlations with CSE ($r = 0.48$) and a moderate negative correlation with the DASS-21 ($r = -0.38$). Connectedness was also negatively correlated with the DASS-21 subscales and had moderate positive correlations with the CSE subscales, as expected. Happiness showed strong correlations with the CSE ($r = 0.61$) and DASS-21 ($r = -0.63$). Moreover, the correlations with the subscales of the DASS-21 were moderate to strong and negative; those with the CSE subscales were moderate to strong and positive.

## Discussion and conclusion

In this study, we investigated the psychometric properties of the Swedish version of the EPOCH Measure of Adolescent Well-being among Swedish high school students (aged 16–21 years). Our results supported the scale's reliability and validity.

The five-factor solution of the origin EPOCH was supported by a CFA, having good overall model fit. In line with previous findings, the five-factor model fit the data better than did the one-factor model or the second-order factor model, even though the latter model also showed acceptable fit. Therefore, both the overall scale scores and subscale scores were used in analyses. Our results showed that the Swedish EPOCH has good internal consistency for overall scores and all five subscales. The particular strengths of the EPOCH are that 1) it is multidimensional in that it takes into account five different dimensions of well-being (engagement, perseverance, optimism, connectedness and happiness) rather than just one, and 2) it accommodates both the hedonic and eudaimonic views of well-being. Considering multiple dimensions of well-being [2,5] as well as both the hedonic and eudaimonic aspects [4–6] gives a more comprehensive picture of one's well-being compared to focusing on just one indicator or aspect.

The subscales also showed good criterion validity. As expected, all five EPOCH subscales had a moderate positive correlation with coping self-efficacy, with engagement showing the lowest correlation, and optimism and happiness showing the highest correlations. Zeng and Kern [31] also found that engagement showed the lowest correlation with other well-being measures. This may be due to the fact that engagement is related to flow, which is a state of complete absorption in an activity without a sense of positive or negative affect [28]. Perhaps feeling engaged is not as strongly related to affectivity as the other EPOCH subscales. Coping

self-efficacy is the belief that one is capable of coping in a challenging circumstance [32]; given that it is a positive self-efficacy belief, it makes sense that optimism and happiness had the strongest relations with it. It is also noteworthy that, as expected, the EPOCH connectedness subscale had a strong correlation with the CSE support from friends and family subscale.

Also as expected, negative correlations were found between the five EPOCH subscales and three indicators of mental health problems, namely depression, anxiety and stress. The weakest negative correlations were found with engagement, and the strongest negative correlations were found with happiness. This, again, replicated the results of Zeng and Kern [31].

The results also indicated scalar invariance across genders, indicating that the five-factor model is sufficient and that mean values are directly comparable across the genders, replicating the findings of Zeng and Kern [31].

## Limitations

The criterion validity was limited because we used only one other positive well-being indicator (coping self-efficacy). We might have included more positive well-being indicators in this study to obtain greater nuance with the criterion validity. However, since the data were based on a larger project, only a couple of comparison indicators were collected. Another limitation with the criterion validity is that all scales were measured in the same way using self-report. Therefore, there is a chance that the criterion validity correlations are inflated due to common method variance [42]. Future validation studies should attempt to use non-self-report measures as well. Furthermore, the data was collected during a global pandemic, which might have affected how students responded to the well-being items. However, we did not consider this a problem for the scale validation, since the levels of well-being at the time of measurement should not interfere with assessment of scale validity or reliability.

## Conclusion

This study indicated that the Swedish version of the EPOCH Measure of Adolescent Well-being had good psychometric properties: the CFA indicated good model fit and the scale (both the total scale and the subscales) had high internal consistency and criterion validity. Therefore, this study provides support for the use of this multidimensional measurement of positive psychological functioning among Swedish-speaking adolescent samples.

## Supporting information

**S1 Appendix. The Swedish version of the EPOCH measure of adolescent well-being.** (DOCX)

## Acknowledgments

We would like to thank Jason Maurer for his editorial expertise.

## Author Contributions

**Conceptualization:** Mia M. Maurer.

**Formal analysis:** Mia M. Maurer, Daiva Daukantaitė.

**Funding acquisition:** Daiva Daukantaitė.

**Supervision:** Daiva Daukantaitė, Eva Hoff.

**Writing – original draft:** Mia M. Maurer.

**Writing – review & editing:** Mia M. Maurer, Daiva Daukantaitė, Eva Hoff.

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
