## [Decision Letter · Decision Letter 0]

19 May 2021

PONE-D-21-08819

Testing the psychometric properties of the Swedish version of the EPOCH measure of adolescent well-being

PLOS ONE

Dear Dr. Maurer,

Thank you for submitting your manuscript to PLOS ONE. After careful consideration, we feel that it has merit but does not fully meet PLOS ONE’s publication criteria as it currently stands. Therefore, we invite you to submit a revised version of the manuscript if you are able to address the points raised during the review process. A part of  the solution might be finding terms better describing what was measured.

We look forward to receiving your revised manuscript.

Kind regards,

Frantisek Sudzina

Academic Editor

PLOS ONE

Journal Requirements:

2. Please specify whether you obtained consent from parents or guardians of the minors who may have participated in this study.

3. Please note that there are several instances where the McDonald’s values are not formatted correctly and therefore appear as a blank box. In your revision, please ensure that these are replaced with the correct symbol.

Thank you for your attention. We look forward to hearing from you.

Reviewers' comments:

Reviewer's Responses to Questions

**Comments to the Author**

1. Is the manuscript technically sound, and do the data support the conclusions?

Reviewer #1: Partly

Reviewer #2: No

2. Has the statistical analysis been performed appropriately and rigorously? 

Reviewer #1: Yes

Reviewer #2: No

3. Have the authors made all data underlying the findings in their manuscript fully available?

Reviewer #1: No

Reviewer #2: No

4. Is the manuscript presented in an intelligible fashion and written in standard English?

Reviewer #1: Yes

Reviewer #2: Yes

5. Review Comments to the Author

Reviewer #1: I reviewed the manuscript entitled "Testing the psychometric properties of the Swedish version of the EPOCH measure of adolescent well-being" for PLOS ONE (PONE-D-21-08819). I read the manuscript word-for-word and have provided comments and suggestions that I hope will improve the manuscript. I am a clinical psychology post-doc at Kent State University with expertise in well-being measurement. My name is David Disabato and I can be contacted by the authors with any questions about this review (ddisab01@gmail.com).

Introduction:

1) The authors refer to the reliability of factors at various points in the introduction section. Reliability refers to observed scores and not latent factors; latent factors have (theoretically) perfect reliability. For example, the authors say on page 5 “They found that the five-factor model was reliable even across different time points, and each factor had high internal consistency”. Instead it should say something like, “The five EPOCH subscale scores were found to have high internal consistency”.

2) I am not sure what the authors mean when they say “… but its consistency over time was low with a range from r = .12 to r = .21.” Consistency can mean different things in psychometrics: How is that term being used here? Are those correlations test-retest coefficients? If yes, how much time had passed between repeat assessments?

3) I thought it was strange that the authors were referring to specific criterion questionnaires in the introduction section. Personally, I think it makes more sense to talk about constructs (e.g., adverse mental health) in the introduction section, and wait to talk about specific measures until the measures section (e.g., DASS-21). Obviously, for a psychometric paper you do talk about the measure of psychometric interest in the introduction. But I think the criterion measures should wait until the methods section.

Methods:

4) I would like to see more information about study recruitment and administration. How were the 852 students from these 3 schools recruited? Was the study encouraged by the school administration? What percentage of the total student body at each school chose to participate (I assume not everyone volunteered)? It says the surveys were distributed via email and completed online. How many students from each school were emailed the online survey – all of them or only a subset? Were they administered at school or at home? How long did it take participants on average to complete the full survey?

5) In the EPOCH part of the measures subsection, the authors should include a sentence telling the reader that the internal consistency of the EPOCH in this sample will be reported in the results section. As the reader, I am used to seeing the internal consistency reported in the measures section and I was confused by it not being there.

6) I would recommend placing the procedures subsection first in the methods section as it lays the foundation for understanding all other aspects of the study methodology. As the reader, I was left confused after the participants section wondering where the information about the procedure was going to be reported.

7) The type II error rate for the chi-square test of model fit is a problem in small samples, not large samples. I think the authors mean to say that the chi-square statistic tends to reject even great fitting models with negligible misfit in large samples.

8) The authors should provide a definition of McDonald’s omega for the reader and how it differs from Cronbach’s alpha as some readers may not be familiar with that index of internal consistency.

Results:

9) The Little MCAR test cannot determine if the data were truly missing completely at random. It can only tell you if the *measures in your dataset* are related to missingness, but that says nothing about the infinite number of other variables not measured in your study. You can certainly report the results of Little’s MCAR test, but I think the authors should replace the statement that says “Thus, the data were missing completely at random” with something like “Thus, the missing values were not related to any variables in our study”.

10) Since there is almost no missing data (<.01%), preliminary analyses about missing data don’t seem very important. Instead, I would like to see correlations, means, and standard deviations of the subscale scores as a Table 1 (as currently a Table 1 does not exist in this paper for some reason). I noticed the authors report the means and standard deviations in Table 2, so I would then move them to this new Table 1. I think it would also be worth including a supplemental table of the EPOCH items correlations, means, and standard deviations as those could be used by other researchers to replicate and extend the CFA analyses the authors have done; however, I will leave that up to the authors.

11) How the authors report the chi-square difference tests of model comparisons is confusing to me. The chi-square difference values appear to look like the alternative model’s chi-square statistic, but then slowly realized that those were the chi-square difference values. I think it would be easier for the reader if the authors first presented the chi-square statistics (and model fit indices) for each model and then follow-up that up with separate sentences about model comparison and the chi-square difference tests.

12) It is not clear what the coefficients are in Figure 1. I assume they are standardized factor loadings and latent correlations, but that needs to be explicitly stated in the caption.

13) If the authors are going to use the EPOCH total score, then they really should be discussing the higher-order factor model as an alternative, acceptably fitting factor model, as that is the factor model behind the total score. Right now, the results and discussion are written as if the higher-order factor model was rejected, which then implies a total score would not be used – only subscale scores. However, that is not what the authors do when moving to criterion validity. Therefore, the authors either need to talk about the higher-order factor model as an alternative, acceptable factor model or remove the analyses with the EPOCH total score. (It does makes sense to me to have an EPOCH total score given the internal consistency of the total score was very high.)

14) The authors don’t need to repeat the Cronbach’s alphas and McDonald’s omegas in Table 2 if they already have them in the text. I would put them in one or the other location, but no need for both.

15) There are some typos in the notes for Table 2.

Discussion:

16) Again, there is psychometric language about factors that should be about subscale scores. For example, on page 13 “The factors also showed good criterion validity”. The criterion validity analyses were done with the EPOCH observed subscale scores, not the latent factors in a full SEM, so the language in the discussion should refer to subscale scores, not latent factors.

17) I am not sure why the authors refer to optimism as a “positive emotion-based subscales” in the discussion section. Optimism is a cognitive construct by almost every definition I have read and the items in the EPOCH all refer to cognitive beliefs and not emotions. They need to either clarify what they mean or remove this statement.

18) The authors should remove the reference to “mood disorders” when talking about the DASS-21 in the discussion on page 14. Both depression and anxiety are sometimes referred to as involving problems with affect or mood, but the term “mood disorder” is reserved to depression and not anxiety in the mental health literature.

19) I find it strange that the authors are interested in showing no gender differences across the EPOCH subscales. Why is that important to the psychometrics? It would be important to show measurement invariance across gender in the factor models, but a gender difference for the subscale scores doesn’t say anything about the psychometrics of the questionnaire. Especially, since there is evidence for gender differences in well-being across measurement methods (e.g., women tend to be higher on both negative emotions and positive emotions; Nolen-Hoeksema & Rusting, 1999).

Nolen-Hoeksema, S., & Rusting, C. L. (1999). Gender Differences in Well-Being. Well-being: Foundations of hedonic psychology, Chapter 17.

20) The same argument goes for age. If the authors are concerned about the EPOCH being psychometrically biased due to gender or age, then they should test for measurement invariance. For example, Kern et al. (2016) tested for measurement invariance across gender. I noticed the authors cite Tim Brown’s CFA book – chapter 7 goes over measurement invariance across demographic groups if they are interested in doing this. Unless they authors test for measurement invariance across age, then they cannot state that “they are invariant with age among adolescents and changes in scores reflect changes in psychological functioning rather than changes in maturation” (pg. 14).

21) I think the authors need to point that that a limitation of the criterion validity tests is that all criterion were measured with the same method as the EPOCH measure (i.e., self-report). Therefore, there is a good chance that the criterion validity correlations are over-estimates of validity and inflated due to shared method variance. Future research should include non-self-report criterion.

Reviewer #2: Thank you for the opportunity to review the manuscript, “Testing the psychometric properties of the Swedish version of the EPOCH measure of adolescent well-being.” The purpose of this manuscript was to translate the EPOCH measure into Swedish. Participants were 846 Swedish adolescents who completed a self-report questionnaire in May 2020.

Strengths: I appreciate research efforts to translate English measures into other languages to increase access to different populations. Psychological science certainly has an issue of recruiting diverse samples, including studies that involve languages other than English. The sample size was sufficiently large and analyses are straightforward.

Nonetheless, this manuscript has noteworthy limitations. These are detailed below.

1) Several broad sweeping assumptions are made in the introduction. The authors describe “hedonia” and “eudaimonia” as two distinct forms of well-being. However, there are now several studies from different research groups showing that these two types of well-being—at least as we currently measure them—overlap quite a high degree, with some people suggesting they represent the same type of well-being. At a minimum, the authors should acknowledge that there is considerable debate about whether or not these represent two distinct forms of well-being and present relevant factor analytical work.

2) Hedonic well-being is described as “emotional well-being” and eudaimonic well-being is described as “well-being resources not tied to emotions.” There are three issues with this conceptualization (in addition to the issue noted above). First, the widely adopted model of hedonic well-being is Diener’s subjective well-being model (SWB), which is composed of the presence of positive emotions, lack of negative emotions, and life satisfaction. Diener and others have amassed a corpus of work demonstrating the utility of this model. (And in the introduction of the present study, the authors use Diener’s model to describe hedonia.) Yet, negative emotions and life satisfaction are surprisingly absent from the EPOCH measurement model. Second, it is incorrect to claim that eudaimonic is “not tied to emotions.” For example, Laura King has shown that meaning in life, arguably a core component of “eudaimonic well-being”—although also surprisingly absent from the EPOCH measure—is correlated with (and often influenced by) positive affect. Can a “type” of well-being really be devoid of or distinct from emotions? Third, the lack of a strong, clear, and cohesive definition of eudaimonic well-being opens up the possibility for any “positive” or desirable psychological variable — as judged, often arbitrarily, by the researcher — to be called “well-being.” For example, why was environmental mastery from Ryff’s widely adopted psychological well-being (PWB) model excluded from the EPOCH model presented here? A 2016 review identified 99 self-report measures of “well-being” containing nearly 200 different dimensions. Without concise definitions, theoretically informed models, and some consistency in measurement, the term “well-being” runs the risk of becoming meaningless.

3) A related but separate point: I am struggling to see how optimism and perseverance are components of well-being. Decades of research would suggest that these are personality traits.

4) P. 4: “However, Kern et al. argued that measuring well-being in adolescents should be based on well-being categories that are less abstract than those in the PERMA profiler (14), which was more suitable for adults.” Can the authors clarify what is meant by “abstract” and elaborate on which constructs were replaced and why? I do not entirely follow the logic.

5) The EPOCH measure is based on the PERMA collection of variables, for which there is limited empirical support. To date, few studies have examined the factor structure of PERMA, and some have found that it highly overlaps with other, more well-established models. Stronger rationale is needed for selection of this variable set.

6) The five components of the EPOCH measure are described as “flourishing” in the results section but “well-being” elsewhere in the manuscript. Please use consistent terminology to prevent confusion.

7) The race/ethnicity breakdown of participants is missing from the demographics section.

8) P. 15: “We might have included more positive well-being indicators in this study to obtain greater nuance with the criterion validity. However, since the data were based on a larger project, only a couple of comparison indicators were collected.” I appreciate the authors being forthcoming in this limitation. However, it does raise significant questions about the validity of this measure. It seems that at a minimum, an assessment of criterion validity should include a comparison with another measure of well-being.

9) I do not see Table 1, only Table 2. Based on the text, it seems that Table 1 would include the bivariate correlations?

6. PLOS authors have the option to publish the peer review history of their article (what does this mean?). If published, this will include your full peer review and any attached files.

Reviewer #1: **Yes: **David Disabato

Reviewer #2: No

---

## [Author Response · Author response to Decision Letter 0]

19 Aug 2021

Journal Requirements:

 Answer 1: We have followed the guidelines now.

2. Please specify whether you obtained consent from parents or guardians of the minors who may have participated in this study.

 Answer 2: All the participants were above the age of 15, meaning that they were able to give their own informed consent without needing parental informed consent under Swedish ethical guidelines. This has now been added into the manuscript. 

3. Please note that there are several instances where the McDonald’s values are not formatted correctly and therefore appear as a blank box. In your revision, please ensure that these are replaced with the correct symbol.

Answer 3: We have made sure that the symbols are in place in the manuscript.

Thank you for your attention. We look forward to hearing from you.

 Answer 4: The ethics committee from which we obtained ethical approval indicated that data should not be shared with outside parties.

Answer 5: We did not obtain ethical approval for sharing data. In our ethical approval, we were told that it is important that we do not share our data with outside parties. However, we want to be transparent and share data with individuals who are interested in it, so we opted to share only upon request. We hope this is acceptable. 

5. Review Comments to the Author

Reviewer #1: I reviewed the manuscript entitled "Testing the psychometric properties of the Swedish version of the EPOCH measure of adolescent well-being" for PLOS ONE (PONE-D-21-08819). I read the manuscript word-for-word and have provided comments and suggestions that I hope will improve the manuscript. I am a clinical psychology post-doc at Kent State University with expertise in well-being measurement. My name is David Disabato and I can be contacted by the authors with any questions about this review (ddisab01@gmail.com).

Introduction:

1) The authors refer to the reliability of factors at various points in the introduction section. Reliability refers to observed scores and not latent factors; latent factors have (theoretically) perfect reliability. For example, the authors say on page 5 “They found that the five-factor model was reliable even across different time points, and each factor had high internal consistency”. Instead it should say something like, “The five EPOCH subscale scores were found to have high internal consistency”.

Answer 6: Thank you for this important observation! This issue has been corrected in the manuscript. 

2) I am not sure what the authors mean when they say “… but its consistency over time was low with a range from r = .12 to r = .21.” Consistency can mean different things in psychometrics: How is that term being used here? Are those correlations test-retest coefficients? If yes, how much time had passed between repeat assessments?

Answer 7: The time between repeat assessments was actually not reported in the original article. These correlations are test-retest coefficients.

3) I thought it was strange that the authors were referring to specific criterion questionnaires in the introduction section. Personally, I think it makes more sense to talk about constructs (e.g., adverse mental health) in the introduction section, and wait to talk about specific measures until the measures section (e.g., DASS-21). Obviously, for a psychometric paper you do talk about the measure of psychometric interest in the introduction. But I think the criterion measures should wait until the methods section.

Answer 8: Thank you for this observation. We have removed the lengthy description of the criterion questionnaires from the introduction section.

Methods:

4) I would like to see more information about study recruitment and administration. How were the 852 students from these 3 schools recruited? Was the study encouraged by the school administration? What percentage of the total student body at each school chose to participate (I assume not everyone volunteered)? It says the surveys were distributed via email and completed online. How many students from each school were emailed the online survey – all of them or only a subset? Were they administered at school or at home? How long did it take participants on average to complete the full survey?

Answer 9: The study link was published on each school’s website, which is accessible to all students. However, we do not know how many of the students read the information about the study. Many students from the school did not participate, which might be due to their not taking notice of the study on the website. We have some statistics on how many of the students clicked the link we posted on the website and how many of them completed the questionnaire. The total number of students who clicked the link was 1212, while only 852 (about 70%) completed the questionnaire. The study was not explicitly encouraged by the school administration, but they did agree to participate. Since the pandemic was ongoing and schools were closed, the questionnaire battery was administered at home. The participation took approximately 15 minutes. Some of these details have now been added to the manuscript for clarity.

5) In the EPOCH part of the measures subsection, the authors should include a sentence telling the reader that the internal consistency of the EPOCH in this sample will be reported in the results section. As the reader, I am used to seeing the internal consistency reported in the measures section and I was confused by it not being there.

Answer 10: A clarification has now been added into the measures section.

6) I would recommend placing the procedures subsection first in the methods section as it lays the foundation for understanding all other aspects of the study methodology. As the reader, I was left confused after the participants section wondering where the information about the procedure was going to be reported.

Answer 11: Thank you for this remark. The placement of the Procedure section has been changed.

7) The type II error rate for the chi-square test of model fit is a problem in small samples, not large samples. I think the authors mean to say that the chi-square statistic tends to reject even great fitting models with negligible misfit in large samples.

Answer 12: Thank you for this remark. This has been corrected. 

8) The authors should provide a definition of McDonald’s omega for the reader and how it differs from Cronbach’s alpha as some readers may not be familiar with that index of internal consistency.

Answer 13: This is an important point, thank you. A clarification of the indexes of internal consistency has been added.

Results:

9) The Little MCAR test cannot determine if the data were truly missing completely at random. It can only tell you if the *measures in your dataset* are related to missingness, but that says nothing about the infinite number of other variables not measured in your study. You can certainly report the results of Little’s MCAR test, but I think the authors should replace the statement that says “Thus, the data were missing completely at random” with something like “Thus, the missing values were not related to any variables in our study”.

Answer 14: Thank you for this important point! This correction has been made in the manuscript.

10) Since there is almost no missing data (<.01%), preliminary analyses about missing data don’t seem very important. Instead, I would like to see correlations, means, and standard deviations of the subscale scores as a Table 1 (as currently a Table 1 does not exist in this paper for some reason). I noticed the authors report the means and standard deviations in Table 2, so I would then move them to this new Table 1. I think it would also be worth including a supplemental table of the EPOCH items correlations, means, and standard deviations as those could be used by other researchers to replicate and extend the CFA analyses the authors have done; however, I will leave that up to the authors.

Answer 15: We have updated our tables and added a new one reporting item intercorrelations and descriptives, as suggested.

11) How the authors report the chi-square difference tests of model comparisons is confusing to me. The chi-square difference values appear to look like the alternative model’s chi-square statistic, but then slowly realized that those were the chi-square difference values. I think it would be easier for the reader if the authors first presented the chi-square statistics (and model fit indices) for each model and then follow-up that up with separate sentences about model comparison and the chi-square difference tests.

Answer 16: Thank you for this comment; this has now been done.

12) It is not clear what the coefficients are in Figure 1. I assume they are standardized factor loadings and latent correlations, but that needs to be explicitly stated in the caption.

Answer 17: These details have now been added to the caption.

13) If the authors are going to use the EPOCH total score, then they really should be discussing the higher-order factor model as an alternative, acceptably fitting factor model, as that is the factor model behind the total score. Right now, the results and discussion are written as if the higher-order factor model was rejected, which then implies a total score would not be used – only subscale scores. However, that is not what the authors do when moving to criterion validity. Therefore, the authors either need to talk about the higher-order factor model as an alternative, acceptable factor model or remove the analyses with the EPOCH total score. (It does makes sense to me to have an EPOCH total score given the internal consistency of the total score was very high.)

Answer 18: Thank you for the comment. Even though the higher-order model had somewhat worse fit compared to the five-factor model, the fit indices were in the acceptable range. We added in the text that the higher-order model was not rejected. 

14) The authors don’t need to repeat the Cronbach’s alphas and McDonald’s omegas in Table 2 if they already have them in the text. I would put them in one or the other location, but no need for both.

Answer 19: These have been removed from the table.

15) There are some typos in the notes for Table 2.

Answer 20: We have had a professional editor check through the manuscript.

Discussion:

16) Again, there is psychometric language about factors that should be about subscale scores. For example, on page 13 “The factors also showed good criterion validity”. The criterion validity analyses were done with the EPOCH observed subscale scores, not the latent factors in a full SEM, so the language in the discussion should refer to subscale scores, not latent factors.

Answer 21: This has now been corrected.

17) I am not sure why the authors refer to optimism as a “positive emotion-based subscales” in the discussion section. Optimism is a cognitive construct by almost every definition I have read and the items in the EPOCH all refer to cognitive beliefs and not emotions. They need to either clarify what they mean or remove this statement.

Answer 22: This is a very important observation. We have omitted the statement.

18) The authors should remove the reference to “mood disorders” when talking about the DASS-21 in the discussion on page 14. Both depression and anxiety are sometimes referred to as involving problems with affect or mood, but the term “mood disorder” is reserved to depression and not anxiety in the mental health literature.

Answer 23: This is a very valid point; accordingly, we have omitted this from the manuscript.

19) I find it strange that the authors are interested in showing no gender differences across the EPOCH subscales. Why is that important to the psychometrics? It would be important to show measurement invariance across gender in the factor models, but a gender difference for the subscale scores doesn’t say anything about the psychometrics of the questionnaire. Especially, since there is evidence for gender differences in well-being across measurement methods (e.g., women tend to be higher on both negative emotions and positive emotions; Nolen-Hoeksema & Rusting, 1999).

Nolen-Hoeksema, S., & Rusting, C. L. (1999). Gender Differences in Well-Being. Well-being: Foundations of hedonic psychology, Chapter 17.

Answer 24: Thank you for this important comment. We tested the measurement invariance across genders focusing on the change in CFI (ΔCFI) and RMSEA (ΔRMSEA) when a more constrained model was compared to a less constrained one. We considered ΔCFI ≤ 0.010 and ΔRMSEA ≤ 0.015 (Cheung & Rensvold, 2002; Chen, 2007) as indications that the invariance assumption holds. The results are now presented in the manuscript.

20) The same argument goes for age. If the authors are concerned about the EPOCH being psychometrically biased due to gender or age, then they should test for measurement invariance. For example, Kern et al. (2016) tested for measurement invariance across gender. I noticed the authors cite Tim Brown’s CFA book – chapter 7 goes over measurement invariance across demographic groups if they are interested in doing this. Unless they authors test for measurement invariance across age, then they cannot state that “they are invariant with age among adolescents and changes in scores reflect changes in psychological functioning rather than changes in maturation” (pg. 14).

Answer 25: We have omitted a comparison of age since our sample is rather homogenous regarding age. 

21) I think the authors need to point that that a limitation of the criterion validity tests is that all criterion were measured with the same method as the EPOCH measure (i.e., self-report). Therefore, there is a good chance that the criterion validity correlations are over-estimates of validity and inflated due to shared method variance. Future research should include non-self-report criterion.

Answer 26: Thank you for pointing out this important limitation, which has been added to the discussion.

Reviewer #2: Thank you for the opportunity to review the manuscript, “Testing the psychometric properties of the Swedish version of the EPOCH measure of adolescent well-being.” The purpose of this manuscript was to translate the EPOCH measure into Swedish. Participants were 846 Swedish adolescents who completed a self-report questionnaire in May 2020.

Strengths: I appreciate research efforts to translate English measures into other languages to increase access to different populations. Psychological science certainly has an issue of recruiting diverse samples, including studies that involve languages other than English. The sample size was sufficiently large and analyses are straightforward.

Answer 27: Thank you for pointing out the strengths of the manuscript!

Nonetheless, this manuscript has noteworthy limitations. These are detailed below.

1) Several broad sweeping assumptions are made in the introduction. The authors describe “hedonia” and “eudaimonia” as two distinct forms of well-being. However, there are now several studies from different research groups showing that these two types of well-being—at least as we currently measure them—overlap quite a high degree, with some people suggesting they represent the same type of well-being. At a minimum, the authors should acknowledge that there is considerable debate about whether or not these represent two distinct forms of well-being and present relevant factor analytical work.

Answer 28: Thank you for this very important observation. It is true that the distinction between hedonic and eudaimonic well-being is under debate, and this should have been brought forward clearly in the manuscript. We have addressed this point by adding more critical discussion of the distinction between these types of well-being and emphasizing the debate in the field.

2) Hedonic well-being is described as “emotional well-being” and eudaimonic well-being is described as “well-being resources not tied to emotions.” There are three issues with this conceptualization (in addition to the issue noted above). First, the widely adopted model of hedonic well-being is Diener’s subjective well-being model (SWB), which is composed of the presence of positive emotions, lack of negative emotions, and life satisfaction. Diener and others have amassed a corpus of work demonstrating the utility of this model. (And in the introduction of the present study, the authors use Diener’s model to describe hedonia.) Yet, negative emotions and life satisfaction are surprisingly absent from the EPOCH measurement model. Second, it is incorrect to claim that eudaimonic is “not tied to emotions.” For example, Laura King has shown that meaning in life, arguably a core component of “eudaimonic well-being”—although also surprisingly absent from the EPOCH measure—is correlated with (and often influenced by) positive affect. Can a “type” of well-being really be devoid of or distinct from emotions? Third, the lack of a strong, clear, and cohesive definition of eudaimonic well-being opens up the possibility for any “positive” or desirable psychological variable — as judged, often arbitrarily, by the researcher — to be called “well-being.” For example, why was environmental mastery from Ryff’s widely adopted psychological well-being (PWB) model excluded from the EPOCH model presented here? A 2016 review identified 99 self-report measures of “well-being” containing nearly 200 different dimensions. Without concise definitions, theoretically informed models, and some consistency in measurement, the term “well-being” runs the risk of becoming meaningless.

Answer 29: Thank you for these important arguments and pointing out the theoretical weaknesses of the manuscript. It is true that we made too broad a generalization concerning hedonic and eudaimonic well-being. Research has indeed shown that these types of well-being are highly interrelated, with some studies showing them to be in fact loading onto just a single overarching factor. We have thus added more nuance in the discussion of these two types of well-being in the introduction and underscored the debate (although still rather briefly). As you note, the EPOCH measure does not include many of the different aspects of other well-being models, such as environmental mastery, autonomy, personal growth, meaning etc. We believe this may be because Kern and colleagues considered those aspects to be too difficult for adolescents to grasp (this is just our guess, as it was not discussed by them) and because the EPOCH was meant to operationalize the PERMA model for adolescents by conceptually simplifying some of its subcomponents, such as meaning and achievement (which became optimism and perseverance, respectively). Our reason for using the EPOCH (which was not clearly stated in the manuscript before, but has been added now) was that the PERMA model is the theoretical framework underlying most positive educational studies. It is therefore important to be able to measure its facets, particularly in the field of positive education. Moreover, many positive educational interventions are based on the PERMA model, and evaluations of the interventions therefore require a measure that operationalizes PERMA for adolescents. This rationale has been more clearly articulated in the manuscript. 

3) A related but separate point: I am struggling to see how optimism and perseverance are components of well-being. Decades of research would suggest that these are personality traits.

Answer 30: This is an apt observation, and a very important discussion point considering the way in which well-being is defined. The EPOCH, although a measure of well-being, is meant to define positive characteristics that predict adolescents’ optimal functioning. As we understand it, Kern and colleagues wanted to create a measure in which not only attitudes, but also characteristics, would be reflected to measure optimal adolescent functioning. In addition, these aspects were meant to reflect the PERMA categories of Achievement (perseverance) and Meaning (optimism) as more action-oriented aspects of functioning. Kern et al. suggest that through perseverance, adolescents are able to reach achievements, while through optimism, they are able to engage with meaningful pursuits and a sense of meaning in life. We have now included more discussion of optimal functioning in the manuscript, and the potential practical reason for the use of personality traits in the well-being definition (to evaluate interventions).

4) P. 4: “However, Kern et al. argued that measuring well-being in adolescents should be based on well-being categories that are less abstract than those in the PERMA profiler (14), which was more suitable for adults.” Can the authors clarify what is meant by “abstract” and elaborate on which constructs were replaced and why? I do not entirely follow the logic.

Answer 31: Pointing out this unclear part of the introduction was important. We decided to delete this part of the manuscript discussing the ‘’abstract’’ PERMA categories, mainly because we could not much elaborate on it – Kern et al. do not elaborate on this point themselves. In its place, we discuss your previous point in more detail – that is, why well-being is operationalized with personality traits. Hopefully, this paragraph is now clearer and explains the EPOCH better.

5) The EPOCH measure is based on the PERMA collection of variables, for which there is limited empirical support. To date, few studies have examined the factor structure of PERMA, and some have found that it highly overlaps with other, more well-established models. Stronger rationale is needed for selection of this variable set.

Answer 32: Thank you for this important observation. We have now added some literature evaluating the psychometric properties of the PERMA profiler, as well as added a clearer rationale for why we chose the PERMA framework for our study. Our rationale for choosing the PERMA framework is two-fold: 1) PERMA is a multidimensional measure of well-being (which does not necessarily set it above or aside many other models, however), and 2) PERMA is the main theoretical framework behind positive education studies, and for theoretical as well as practical purposes (such as intervention evaluations) it is important to validate a measure of PERMA for adolescents. 

6) The five components of the EPOCH measure are described as “flourishing” in the results section but “well-being” elsewhere in the manuscript. Please use consistent terminology to prevent confusion.

Answer 33: Thank you for pointing this out. We have corrected this confusion now.

7) The race/ethnicity breakdown of participants is missing from the demographics section.

Answer 34: Thank you for pointing this out. We acknowledge the lack of such information in the participant description. For ethical reasons, however, it is not customary in Sweden to ask participants for race/ethnicity information unless there is a very important reason to do so. We did not have such a reason, and therefore we did not include this question in the demographic questions of the study. This is a limitation, unfortunately. However, in Sweden, it is acceptable to ask which country the students are from and which country their parents are from. Based on this information, we identify a group of participants with a foreign background. This is defined as the participant either being born abroad with at least one parent born abroad as well, or being born in Sweden with both parents being born abroad. This information was included in the Methods.

8) P. 15: “We might have included more positive well-being indicators in this study to obtain greater nuance with the criterion validity. However, since the data were based on a larger project, only a couple of comparison indicators were collected.” I appreciate the authors being forthcoming in this limitation. However, it does raise significant questions about the validity of this measure. It seems that at a minimum, an assessment of criterion validity should include a comparison with another measure of well-being.

Answer 35: We acknowledge this serious limitation in our study. It is unfortunate that we did not include another measure of well-being. We have only one measure reflecting positive psychological functioning, the coping self-efficacy scale, which we hope is sufficient for criterion validity. However, we understand that this might raise questions of the EPOCH’s validity.

9) I do not see Table 1, only Table 2. Based on the text, it seems that Table 1 would include the bivariate correlations?

Answer 36: Now we have included Table 1, and two additional tables into the manuscript.

---

## [Editor Report · Decision Letter 1]

15 Oct 2021

Testing the psychometric properties of the Swedish version of the EPOCH measure of adolescent well-being

PONE-D-21-08819R1

Dear Dr. Maurer,

We’re pleased to inform you that your manuscript has been judged scientifically suitable for publication and will be formally accepted for publication once it meets all outstanding technical requirements.

Kind regards,

Frantisek Sudzina

Academic Editor

PLOS ONE
---

## [Editor Report · Acceptance letter]

20 Oct 2021

PONE-D-21-08819R1 

Testing the psychometric properties of the Swedish version of the EPOCH measure of adolescent well-being 

Dear Dr. Maurer:

I'm pleased to inform you that your manuscript has been deemed suitable for publication in PLOS ONE. Congratulations! Your manuscript is now with our production department. 

Kind regards, 

on behalf of

Dr. Frantisek Sudzina 

Academic Editor

PLOS ONE